# Influence of Dexamethasone on the Plasma and Milk Disposition Kinetics of Danofloxacin in Lactating Sheep

**DOI:** 10.3390/vetsci12030210

**Published:** 2025-03-01

**Authors:** Kamil Uney, Ramazan Yildiz, Duygu Durna Corum, Merve Ider, Orhan Corum

**Affiliations:** 1Department of Pharmacology and Toxicology, Faculty of Veterinary Medicine, University of Selcuk, Konya 42031, Türkiye; 2Department of Internal Medicine, Faculty of Veterinary Medicine, University of Burdur Mehmet Akif Ersoy, Burdur 15030, Türkiye; ramazanyildiz@mehmetakif.edu.tr; 3Department of Pharmacology and Toxicology, Faculty of Veterinary Medicine, University of Hatay Mustafa Kemal, Hatay 31000, Türkiye; ddurnacorum@gmail.com (D.D.C.); orhancorum46@hotmail.com (O.C.); 4Department of Internal Medicine, Faculty of Veterinary Medicine, University of Selcuk, Konya 42100, Türkiye; m.ider@selcuk.edu.tr

**Keywords:** danofloxacin, dexamethasone, drug interactions, milk, pharmacokinetics, sheep

## Abstract

This study examines the effects of different doses of dexamethasone on the behavior of danofloxacin in the plasma and milk of lactating sheep. Understanding this interaction is crucial as both drugs are frequently used in veterinary medicine and may influence therapeutic effectiveness and milk safety. The research involved six lactating ewes, with plasma and milk samples collected at various intervals after administering low and high doses of dexamethasone alongside danofloxacin. The results revealed that although dexamethasone did not alter danofloxacin disposition in the plasma, it significantly increased its concentration in the milk, particularly at higher doses. This raises important concerns for both therapeutic efficacy and milk safety, as it indicates that milk from sheep treated with dexamethasone could contain higher danofloxacin concentrations. Overall, the findings underscore the need for careful evaluation of these medications’ combined use to ensure both optimal treatment outcomes for animals and the safety of milk products for consumers.

## 1. Introduction

The first quinolone, nalidixic acid, was modified to enhance its absorption and efficacy by adding a fluorine molecule, resulting in a fluoroquinolone. Fluoroquinolone antibiotics are classified into generations based on their chemical structure and activity [1]. Danofloxacin is a second-generation fluoroquinolone antibiotic approved for veterinary use. It acts as a bactericidal agent by inhibiting the bacterial DNA gyrase enzyme [2,3]. Danofloxacin is active against a wide range of bacteria, including Gram-negative, Gram-positive, and *Mycoplasma* species [4,5]. Danofloxacin has a longer elimination half-life, a greater volume of distribution, and higher oral bioavailability than other fluoroquinolone antibiotics, such as enrofloxacin and marbofloxacin [6,7]. Danofloxacin has been approved for use in respiratory system infections in cattle, pigs, and chickens [8]. It is also recommended for use in diseases in the central nervous system and urinary system, as well as in soft tissue infections, the prostate, osteomyelitis, enteritis, otitis, and ophthalmitis [9,10]. Although danofloxacin is not licensed for use in sheep, it is used in an extra-label way for the disorders mentioned above. Glucocorticoids are 21-carbon steroid hormones synthesized in the cortex of the adrenal gland [11]. Glucocorticoids affect carbohydrate, fat, and protein metabolism, the cardiovascular system, fluid and electrolyte balance, inflammation and the immune system, reproduction, mood, and embryonic and cognitive development [12]. Dexamethasone is a synthetic glucocorticoid with limited mineralocorticoid activity. It is used in various diseases for its anti-inflammatory, immunosuppressive, and anti-allergic effects [13,14]. In animals, it is utilized for sepsis/endotoxemia, edema, metabolic inflammatory conditions, rheumatic, allergy, and dermatological diseases, acute mastitis, furuncles, burns, poisoning, shock, primary ketosis, labor induction, and joint and tendon problems [15,16]. The effects of dexamethasone are dose-dependent; in sheep, it has been used at doses ranging from 0.05 to 1 mg/kg, with higher doses indicated for shock [17].

Fluoroquinolone antibiotics and dexamethasone are used in combination for bacterial infections [16,18,19]. The concurrent administration of drugs may lead to pharmacokinetic drug interactions. Pharmacokinetic interactions may vary depending on the drug and animal species [20,21,22]. The antibacterial efficacy of danofloxacin is contingent upon concentration and variations in its pharmacokinetics may modify its therapeutic impact. Failure to treat sheep with danofloxacin results in economic losses owing to diseases and the development of bacterial resistance, which is a significant public health issue. There are many studies demonstrating the change in the pharmacokinetics of danofloxacin in combined drug administration [23,24,25,26]. Although there are many studies showing the changes in the pharmacokinetics of danofloxacin in combined drug use, no study has been found on the effect of dexamethasone. The aim of this study was to determine the effects of low (0.1 mg/kg) and high (1 mg/kg) doses of dexamethasone administration on plasma and milk pharmacokinetics after intravenous (6 mg/kg) administration of danofloxacin.

## 2. Materials and Methods

### 2.1. Animals

The experiment included six healthy lactating Akkaraman sheep (milk production 335 ± 25 g/day), averaging 42.57 kg in weight and aged between 2.4 and 2.8 years. The study was conducted on the 30–70 days of lactation. Before the investigation, the health condition of the sheep was verified by physical examination and laboratory tests, including hematological and biochemical assessments. Udder health was assessed using the California mastitis test and udder palpation. They were confined in enclosures with straw bedding and received ad libitum access to water and food. For a minimum period of 20 days prior to the initiation of the trial, the animals were administered a diet devoid of any pharmacological substances. Numbered collars were used to easily recognize the sheep. The Selcuk University Faculty of Veterinary Medicine Ethics Committee approved (approval number: 2019/05) all animal-related treatments.

### 2.2. Experimental Design

The IV jugular catheter (22 G, 0.9 25 mm) was placed under aseptic conditions. The first catheter (right) was utilized for the administration of medications, while the second catheter (left) was employed for sample collection. Sheep body weights were taken one hour before giving the medication to determine the suitable dose. Danofloxacin was administered IV at a dose of 6 mg/kg at all periods. Dexamethasone was also used in low (0.1 mg/kg) and high (1 mg/kg) doses via the same route of administration. A three-period cross-over trial (2 × 2 × 2) was designed, with a minimum of a 20-day washout interval between each period. Six sheep were separated into three subgroups, with two animals in each group. In the first stage, danofloxacin, danofloxacin+low-dose dexamethasone, and danofloxacin+high-dose dexamethasone were applied to the subgroups, respectively. In the second and third stages, the groups were changed to receive different treatments, and at the end of the study, all 6 animals received all treatments. Blood samples (2 mL) were taken at 18 different times (0, 0.08, 0.17, 0.25, 0.5, 0.75, 1, 2, 4, 6, 8, 10, 12, 18, 24, 36, and 48 h) after danofloxacin administration in all groups. Blood was collected by catheter for the first 12 h and by the venipuncture method at other times and collected into tubes containing heparin. Blood samples were centrifuged (4000× *g* for 10 min) within one hour and plasma samples were obtained. Milk samples (2 mL) were collected at the aforementioned times designated for blood samples. Plasma and milk samples were stored at −80 °C and danofloxacin analysis was performed within 3 months. Furthermore, animals were monitored for adverse effects during the investigation.

### 2.3. Analytical Methods

Danofloxacin (≥98.0%, Sigma-Aldrich, St. Louis, MO, USA) was measured using an HPLC assay with an ultraviolet detector. The HPLC apparatus (Shimadzu, Shimadzu, Tokyo, Japan) utilized was similar to that previously described [27,28]. Briefly, 200 µL of plasma and milk samples were added into amber microcentrifuge tubes. Proteins were denatured by adding 400 μL of acetonitrile. The mixture was vortexed (10 min) and centrifuged (10,000× *g* for 10 min). A total of 100 µL of the upper phase was taken and transferred to a new amber microcentrifuge tube, and 100 µL of ultrapure water was added. The mixture was vortexed for 5 s, then transferred to vials, and 10 µL was injected into the HPLC system. The chromatographic separation of danofloxacin was conducted at a wavelength of 280 nm utilizing an inertSustain C18 analytical column (250 × 4.6 mm; 5 μm particle size). The mobile phase with a flow rate of 1 mL/min consisted of 0.4% triethylamine + 0.4% orthophosphoric acid (82%) and acetonitrile (18%).

### 2.4. Method Validation

Danofloxacin stock solutions were diluted (200 μg/mL) in ultrapure water and stored at −80 °C. To establish working standards (0.04–10 μg/mL) for plasma and milk, a stock solution was diluted. To obtain calibration standards for plasma (0.04–10 μg/mL) and milk (0.04–40 μg/mL), working standards and stock solutions were diluted. The method of analysis exhibited remarkable linearity (R^2^ > 0.9989) for calibration standards of milk and plasma. Employing the same equipment and operator, high (10 μg/mL), medium (1 μg/mL), and low (0.1 μg/mL) calibration concentration standards were tested six times daily over three distinct days to assess precision, accuracy, and recovery. The recoveries of danofloxacin were assessed using detector responses from quality control samples and working standards. The recovery ratio was determined to be greater than 87%. The limit of detection and lower limit of quantification were 0.02 and 0.04 μg/mL, respectively. Intra-day and inter-day precision demonstrated values lower than 7.42% and 8.20%, respectively. Accuracy varied from −4.2% to 6.6%.

### 2.5. Pharmacokinetic Analysis

Plasma concentration time curves of danofloxacin were drawn for each sheep. The suitable pharmacokinetic model was identified by visual analysis of individual concentration–time curves and the application of Akaike’s Information Criterion. The pharmacokinetic parameters were assessed by non-compartmental analysis utilizing WinNonlin 6.1.0.173 software (Pharsight, Certara, St. Louis, MO, USA). The definitions and abbreviations of each pharmacokinetic parameter are provided in the footnote of Table 1. The penetration of danofloxacin into milk was determined by the area under the concentration–time curve (AUC)_0–∞ milk_/AUC_0–∞ plasma_ ratio.

### 2.6. Statistical Analysis

The T_max_ was reported as the median (minimum–maximum), whereas other pharmacokinetic parameters were expressed as the geometric mean (minimum–maximum). Statistical analysis was conducted with the SPSS 22.0 software (IBM Corp., Armonk, NY, USA). The one-way analysis of variance and post hoc Tukey test were used to examine the statistical differences across treatment groups. The value of *p* < 0.05 was recognized as the threshold for statistical significance.

## 3. Results

### 3.1. Safety

The administration of danofloxacin to sheep, either alone or in combination with low or high doses of dexamethasone, did not result in any clinical adverse effects. The behavior, movements, appetite, defecation, and urination of the animals were normal throughout the study.

### 3.2. Plasma Pharmacokinetic Parameters

Plasma concentration–time curves and pharmacokinetic parameters either alone or with simultaneous administration of danofloxacin with low (0.1 mg/kg) and high (1 mg/kg) doses of dexamethasone in sheep are presented in Figure 1 and Table 1, respectively. Danofloxacin was detected in plasma up to 24 h in both the alone and combined groups. Following IV administration of danofloxacin at a dose of 6 mg/kg, plasma t_1/2ʎz_, MRT_0–∞_, AUC_0–∞,_ Cl_T_, and V_dss_ values were 5.20 h, 6.53 h, 9.26 h*µg/mL, 0.65 L/h/kg, and 4.23 L/kg, respectively. The administration of both low and high dosages of dexamethasone did not result in any variations in the plasma t_1/2ʎz_, MRT_0–∞_, AUC_0–∞_, Cl_T_, and V_dss_ parameters of danofloxacin.

### 3.3. Milk Pharmacokinetic Parameters

Milk concentration–time curves and pharmacokinetic parameters either alone or with simultaneous administration of danofloxacin with low (0.1 mg/kg) and high (1 mg/kg) doses of dexamethasone in sheep are presented in Figure 2 and Table 1, respectively. Danofloxacin was detected in milk up to 36 h in both the alone and combined groups. Following danofloxacin administration, milk t_1/2ʎz_, AUC_0–∞_ and C_max_ values in milk were 4.30 h, 99.52 h*µg/mL, and 20.61 µg/mL, respectively. Low- and high-dose dexamethasone prolonged milk t_1/2ʎz_. High-dose dexamethasone administration significantly increased the AUC_0–∞_ and C_max_ of danofloxacin in milk. Following the injection of danofloxacin alone, the AUC_0–∞ milk_/AUC_0–∞ plasma_ ratio was 10.75. Low- and high-dose dexamethasone administration increased the AUC_0–∞ milk_/AUC_0–∞ plasma_ ratio.

## 4. Discussion

The concurrent administration of medications may result in pharmacokinetic drug interactions. Pharmacokinetic drug interactions may alter medication effects, leading to treatment failure or adverse effects. The ineffectiveness of antibiotic therapy might result in the emergence of bacterial resistance, therefore severely restricting the application of antibiotics. Danofloxacin and dexamethasone can be used together for bacterial infections such as mastitis and pneumonia. There were no studies that investigated the pharmacokinetic interactions of danofloxacin and dexamethasone when administered concurrently. This study established for the first time the influence of dexamethasone on danofloxacin plasma and milk pharmacokinetics. This study demonstrated that dexamethasone modifies the milk pharmacokinetics of danofloxacin in a dose-dependent manner in sheep.

The concurrent treatment of danofloxacin with both low and high doses of dexamethasone in sheep did not result in any clinically adverse effects. The single administration of danofloxacin to sheep [29,30] and the combined administration of dexamethasone with enrofloxacin to pigs were well tolerated [31]. The recommended dose range in sheep is 1.25–6 mg/kg for danofloxacin [23,32] and 0.04–1 mg/kg for dexamethasone [17,33]. However, as the 1.25 mg/kg dosage is unsuccessful for some diseases in sheep, the use of a 6 mg/kg dose is suggested [25,32]. The effect of dexamethasone is dose-dependent, and high doses are recommended in cases of severe infections and shock [17]. The intravenous administration of both drugs was preferred because it is the recommended route of administration and bioavailability would not affect the pharmacokinetics of danofloxacin.

Following IV injection of danofloxacin to sheep, the t_1/2ʎz_, Cl_T_, and V_dss_ were 5.20 h, 0.65 L/h/kg, and 4.23 L/kg, respectively. Danofloxacin administered to sheep via the same route at doses of 1.25 and 6 mg/kg resulted in t_1/2ʎz_, Cl_T_, and V_dss_ values of 2.08–3.39 h, 0.63–0.79 L/h/kg, and 1.9–2.76 L/kg [29,30,32,34]. Although the Cl_T_ in this study aligned with previous findings, the t_1/2ʎz_ and V_dss_ exhibited discrepancies. The t_1/2ʎz_ is a hybrid parameter consisting of Cl_T_ and V_dss_ and is directly proportional to V_dss_ [35]. In this study, the long t_1/2ʎz_ may be due to the large V_dss_. The volume of distribution varies based on the drug’s physicochemical qualities, the ratio of binding to plasma proteins, and the organism’s body composition [36]. Danofloxacin has a large volume of distribution due to its lipophilicity and low (13–36%) binding to plasma proteins [27,30]. The large volume of distribution allows high penetration into the cell and good activity against susceptible intracellular pathogens [30]. The difference in V_dss_ of danofloxacin in sheep may be due to differences in breed, physiological state, lactation period, dose, weight, and analysis method.

Drugs are frequently excreted into milk by passive diffusion. The molecular weight, ionization, lipophilicity, and protein binding affinity are critical factors influencing medication entry into milk via passive diffusion [37,38]. Danofloxacin is amphoteric and classified as zwitterionic due to the presence of a carboxylic acid and a basic amine functional group. Nonetheless, within the pH range of 6 to 8, these chemicals have enough lipid solubility to permeate tissues [9]. The milk pH of sheep is 6.6 to 6.8 and danofloxacin passes well into milk [29]. Moreover, danofloxacin is a substrate of the breast cancer resistance protein (BCRP), which is crucial for its transfer into milk. BCRP is an efflux protein belonging to the ATP-binding cassette transporter family and facilitates the transfer of medicines from plasma to milk [26,39]. The AUC_milk_/AUC_plasma_ ratio is frequently employed to quantify the degree of drug transfer into milk. A ratio over 1 signifies that the drug accumulates in milk [40]. The AUC_0–∞ milk_/AUC_0–∞ plasma_ ratio after IV administration of danofloxacin to lactating sheep was 10.75. Similar AUC_0–∞ milk_/AUC_0–∞ plasma_ ratios (9.58–9.63) were reported in previous studies in sheep [25,26,39]. This shows that danofloxacin passes into milk at very high levels.

The administration of both low and high dosages of dexamethasone to sheep did not result in any variations in the plasma pharmacokinetics of danofloxacin. Similarly, the plasma pharmacokinetics of danofloxacin were unaffected by the administration of ivermectin, triclabendazole, and isoflavones to sheep [25,26,39]. However, a meloxicam research study in sheep and another ivermectin investigation revealed notable alterations in plasma pharmacokinetics of danofloxacin [23,24]. Dexamethasone decreased the plasma concentration and increased the Cl_T_ of phenytoin, triazolam, and oseltamivir when administered concomitantly in humans [41,42,43]. Danofloxacin is minimally metabolized in the liver and subsequently excreted via urine and bile [44]. The urinary concentration of danofloxacin is 2–3 times higher than the feces concentration [45]. Dexamethasone is excreted in urine and bile after being significantly metabolized in the liver. Metabolism is important in its elimination, and renal excretion accounts for 10% of total clearance [46]. Danofloxacin is a substrate of P-glycoprotein (P-gp), multidrug resistance-associated protein (MRP) 2, and BCRP, while dexamethasone is an inductor of these carrier proteins [24,47]. The reason why dexamethasone administration did not affect the plasma pharmacokinetics of danofloxacin in sheep may be due to different elimination pathways or interactions between multiple transporters.

The administration of high-dose dexamethasone to sheep elevated the AUC_0–∞_ and C_max_ of danofloxacin in milk from 99.52 to 129.95 h*µg/mL and from 20.61 to 24.93 µg/mL, respectively. Furthermore, both low and high doses of dexamethasone extended t_1/2ʎz_ from 4.30 to 4.65–4.85 h. Danofloxacin is a substrate for P-gp, MRP-2, and BCRP transport, but BCRP is important for its excretion into milk because P-gp and MRP-2 are not significantly expressed in lactating mammary glands [39]. BCRP is a crucial determinant in the secretion of danofloxacin into milk, as evidenced by the nearly twofold greater milk-to-plasma ratio in wild-type mice compared to Bcrp1 knockout mice [26]. The activation or inhibition of BCRP modifies the excretion of danofloxacin in milk. Ivermectin and a soy-enriched diet in sheep decreased danofloxacin excretion into milk by 40% and 50%, respectively, owing to BCRP inhibition [25,26]. However, in sheep, the BCRP inhibitor triclobendazole did not change the milk pharmacokinetics of danofloxacin; the possible reason for this was shown to be the low milk concentration of triclobendazole [39]. Dexamethasone passes into milk in cattle at a ratio of 40% plasma [48]. The induction of BCRP may enhance the excretion of its substrates in milk. The milk AUC_0–∞_ (30%) and C_max_ (21%) of danofloxacin may have increased due to the induction of BCRP by high-dose dexamethasone in sheep. The injection of low-dose dexamethasone elevated the milk AUC_0–∞_ and C_max_, although this rise was statistically insignificant. The inducing effect of dexamethasone on BCRP is dose-dependent [47], and the lack of difference in the low-dose group may be due to insufficient dose. BCRP is found in tissues such as the mammary gland as well as the intestine, liver, and kidney [25]. BCRP is highly expressed in the mammary gland during lactation in sheep [49]. Dexamethasone modified the milk pharmacokinetics of danofloxacin, although it did not affect plasma pharmacokinetics. This may be due to their higher sensitivity to dexamethasone due to the high expression of BCRPs in the mammary gland.

The change in the passage of drugs into milk is clinically and toxicologically important [26]. The penetration of antibiotics into the mammary gland is important in the systemic treatment of mastitis [30]. Danofloxacin milk concentrations that are roughly ten times greater than plasma concentrations may offer a benefit in the treatment of mastitis. Dexamethasone can be used in combination with danofloxacin in the treatment of mastitis due to its strong anti-inflammatory properties [16]. Since the antibacterial effect of danofloxacin is concentration-dependent, changes in its concentration may also cause changes in its efficacy [7]. The AUC_0–∞ milk_/AUC_0–∞ plasma_ ratio of danofloxacin in sheep was elevated by the administration of low and high doses of dexamethasone. Therefore, combined administration of dexamethasone may increase the antibacterial efficacy of danofloxacin. The passage of drugs into milk also has toxicological significance. Increased passage of danofloxacin into milk may cause a residue risk when used to treat infections other than mastitis [26]. The presence of antibiotic residues may cause undesirable effects and the development of bacterial resistance [50]. Fluoroquinolones are crucial antibiotics in terms of antimicrobial resistance risk and are classified as “highest priority critically important antimicrobials” by the World Health Organization and “category B (restriction)” by the European Medicines Agency. Therefore, it is essential to pay attention to the withdrawal time to prevent the development of resistance to these antibiotics [51,52]. The withdrawal time of drugs in foods derived from animals should be taken into consideration in order to protect the public’s health and reduce the likelihood of exposure to drug residues originating from food items. It should be kept in mind that the duration of drug withdrawal time may change in combined use, as dexamethasone increases the passage of danofloxacin into milk.

## 5. Conclusions

This study offers important insights into the pharmacokinetics of danofloxacin in lactating sheep and the effects of co-administering dexamethasone. Although neither low nor high doses of dexamethasone altered the plasma pharmacokinetics of danofloxacin, notable changes were observed in its milk pharmacokinetic profile, particularly with high-dose dexamethasone, which resulted in increased AUC and C_max_ values. The significant accumulation of danofloxacin in milk, as reflected by the increased milk-to-plasma AUC ratio, emphasizes the necessity of considering drug interactions and their potential impact on milk safety. These findings underscore the need for diligent oversight in drug selection and administration in lactating sheep to maintain therapeutic efficacy and adhere to food safety regulations. Additionally, further research is essential to develop clear guidelines for the concurrent use of dexamethasone and danofloxacin in veterinary practice.

## Figures and Tables

**Figure 1 vetsci-12-00210-f001:**
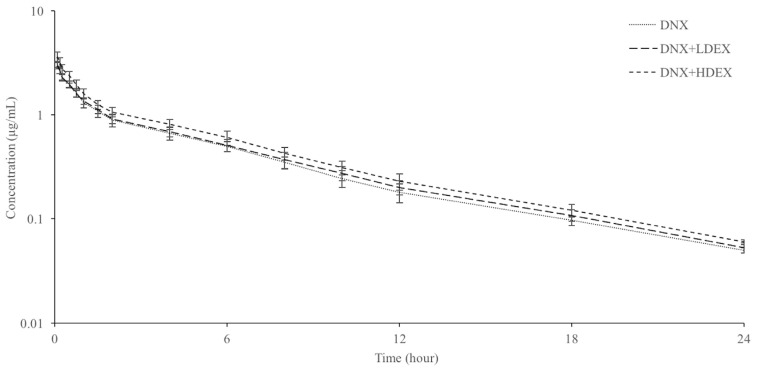
Semi-logarithmic plasma concentration–time curves of danofloxacin (DNX) after intravenous injection (6 mg/kg) alone and co-administered with low (L) and high (H) doses of dexamethasone (DEX) in sheep (n = 6).

**Figure 2 vetsci-12-00210-f002:**
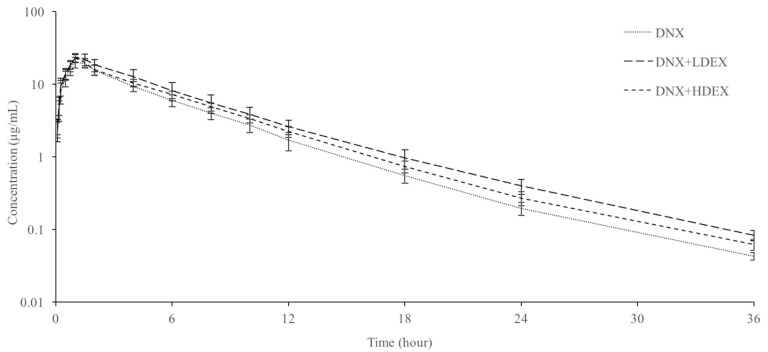
Semi-logarithmic milk concentration–time curves of danofloxacin (DNX) after intravenous injection (6 mg/kg) alone and co-administered with low (L) and high (H) doses of dexamethasone (DEX) in sheep (n = 6).

**Table 1 vetsci-12-00210-t001:** Pharmacokinetic parameters of danofloxacin (DNX) after intravenous injection (6 mg/kg) alone and co-administered with low (L) and high (H) doses of dexamethasone (DEX) in sheep (n = 6).

Parameter	DNX	DNX + LDEX	DNX + HDEX
**Plasma**			
t_1/2ʎz_ (h)	5.20 (4.95–5.41)	5.30 (5.09–5.53)	5.36 (5.06–5.66)
AUC_0–24_ (h*µg/mL)	8.88 (8.31–10.62)	8.62 (7.60–9.85)	9.30 (8.35–10.76)
AUC_0–∞_ (h*µg/mL)	9.26 (8.69–11.01)	8.99 (7.97–10.23)	9.71 (8.74–11.22)
AUC_extrap_ (%)	4.03 (3.50–4.55)	4.20 (3.89–4.54)	4.06 (3.46–4.64)
MRT_0–∞_ (h)	6.53 (6.06–7.00)	6.65 (6.43–6.89)	6.79 (6.58–7.00)
Cl_T_ (L/h/kg)	0.65 (0.55–0.69)	0.67 (0.59–0.75)	0.62 (0.53–0.69)
V_dss_ (L/kg)	4.23 (3.53–4.83)	4.44 (3.78–5.19)	4.20 (3.70–4.56)
C_0.08 h_ (µg/mL)	3.06 (2.80–3.40)	2.88 (2.46–3.38)	2.97 (2.68–3.24)
**Milk**			
t_1/2ʎz_ (h)	4.30 (4.08–4.55) ^b^	4.65 (4.44–4.98) ^a^	4.85 (4.58–5.15) ^a^
AUC_0–24_ (h*µg/mL)	98.04 (78.65–117.44) ^b^	111.96 (92.13–129.92) ^ab^	127.00 (97.50–165.83) ^a^
AUC_0–last_ (h*µg/mL)	99.25 (79.64–118.81) ^b^	113.64 (93.44–131.95) ^ab^	129.37 (99.15–153.87) ^a^
AUC_0–∞_ (h*µg/mL)	99.52 (79.85–119.10) ^b^	114.06 (93.73–132.42) ^ab^	129.95 (99.65–153.87) ^a^
AUC_extrap_ (%)	0.27 (0.23–0.37)	0.44 (0.35–0.53)	0.36 (0.31–0.44)
C_max_ (µg/mL)	20.61 (17.94–24.15) ^b^	23.74 (19.84–28.26) ^ab^	24.93 (21.39–28.04) ^a^
T_max_ (h)	1.00 (1.00–1.50)	1.00 (1.00–1.50)	1.00 (1.00–1.50)
AUC_0–∞ milk_/AUC_0–∞ plasma_	10.75 (9.13–13.11) ^b^	12.69 (9.16–16.20) ^a^	13.38 (9.85–19.38) ^a^

Note: Data were presented as the geometric mean (min–max) except for T_max_, which was presented as the median (min–max). ^a,b^: Various letters in the same row are statistically significant (*p* < 0.05). AUC, area under the concentration–time curve; AUC_extrap_ %, area under the concentration–time curve extrapolated from t_last_ to ∞ in % of the total AUC; C_0.08 h_, plasma concentration at time 0.08 h; Cl_T_, total body clearance; C_max_, peak plasma concentration; MRT_0–∞_, mean residence time; t_1/2λz_, terminal elimination half-life; T_max_, time to reach peak plasma concentration; V_dss_, volume of distribution at steady state.

## Data Availability

The data presented in this study are available upon request from the corresponding author.

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
