# Peer review of "Influence of Dexamethasone on the Plasma and Milk Disposition Kinetics of Danofloxacin in Lactating Sheep"

_vetsci, 2025, doi:10.3390/vetsci12030210_

Round 1
Reviewer 1 Report
Comments and Suggestions for Authors
General comments:
The manuscript deals with an important subject focused on the drug-drug interaction by analyzing the influence of dexamethasone on danofloxacin PK using plasma and milk sheep samples.
Study design, ethical concerns (Ethics Committee approval), analytical methodology and validation data were provided, and pharmacological explanation of the interaction was assayed.
This interaction was not addressed before, thus may be useful in the context of extra-label treatments in sheep.
Findings underscored the impact of combined drug use of danofloxacin and dexamethasone on efficacy and safety, including food safety.
Some minor issues to be clarified/ameliorated:
Line 49-87 (Introduction)
- Improvements in English and edition are recommended throughout, specially in the two first introduction paragraphs.
In: “In animals, it is utilized for sepsis/endotoxemia, edema, metabolic inflammatory conditions, rheumatic, allergy, and dermatological diseases, acute, mastitis, furuncles, burns, poisoning, shock, primary ketosis, labor induction, and join and tendon problems [15,16]”
- This should be rephased and references (15 and 16) should be replaced with several references which adequately support each clinical indication.
Line 66
“Glucocorticoids have effects on carbohydrate, fat, and protein metabolism, cardiovascular system, fluid and electrolyte balance, inflammation and immune system, reproduction and embryonic development, mood and cognitive [12].
- Suggestion to rephase: “…mood, embryonic and cognitive development?
Line 113
Blood samples were collected (with heparin) at 18 different times (0, 0.08, 0.17, 0.25, 0.5, 0.75, 1, 2, 4, 6, 8, 10, 12, 18, 24, 36, and 48 h) and plasma and milk samples were adequately stored at -80 °C until analysis. Validation data is in accordance with criteria.
- Nevertheless, neither the storage period nor stability of drug was assayed. Please comment.
- Was photodegradation considered? (protected from light?)
Line 151
“The definitions and abbreviations of each pharmacokinetic parameter are provided in the footnote of Table 1.”
- This is not necessary.
Line 198
In table legend: “a,b,c: Various letters in the same row are statistically significant (p<0.05)”.
- Regarding the statistical significance, clarification of the use “a, b, c” would be helpful.
Line 280
“However, in sheep, the BCRP inhibitor triclobendazole metabolites did not change the milk pharmacokinetics of danofloxacin;”
- Please consider “triclabendazole” (two items found as triclobendazole)
Line 305
Considering resistance global problems and knowing that fluoroquinolones are categorized by WHO as “highest priority critically important antimicrobials” (HPCIA) and by EMA (AMEG category – cat B “Restrit”), further discussion/reference to this topic would be advisable/welcome.
Comments on the Quality of English Language
Regarding "Introduction", improvements in English and research background are recommended.
Author Response
Reviewer 1
General comments:
The manuscript deals with an important subject focused on the drug-drug interaction by analyzing the influence of dexamethasone on danofloxacin PK using plasma and milk sheep samples.
Study design, ethical concerns (Ethics Committee approval), analytical methodology and validation data were provided, and pharmacological explanation of the interaction was assayed.
This interaction was not addressed before, thus may be useful in the context of extra-label treatments in sheep.
Findings underscored the impact of combined drug use of danofloxacin and dexamethasone on efficacy and safety, including food safety.
Response: Thank you for your valuable opinions and contributions.
Some minor issues to be clarified/ameliorated:
Line 49-87 (Introduction)
- Improvements in English and edition are recommended throughout, specially in the two first introduction paragraphs.
- Response: English of the relevant parts has been revised.
In: “In animals, it is utilized for sepsis/endotoxemia, edema, metabolic inflammatory conditions, rheumatic, allergy, and dermatological diseases, acute, mastitis, furuncles, burns, poisoning, shock, primary ketosis, labor induction, and join and tendon problems [15,16]”
- This should be rephased and references (15 and 16) should be replaced with several references which adequately support each clinical indication.
- Response: The relevant references have been replaced with the following references.
- Yahi, D., Nggada, H. A., Ojo, N. A., Mahre, M. B., Igbokwe, N. A., Umaru, B., & Mshelia, G. D. (2017). Dexamethasone uses in humans and animals: public health and socio-economic implications. Kanem Journal of Medical Sciences, 11(2), 60-67.
- Aharon MA, Prittie JE, Buriko K. A review of associated controversies surrounding glucocorticoid use in veterinary emergency and critical care. J Vet Emerg Crit Care (San Antonio). 2017 May;27(3):267-277.
Line 66
“Glucocorticoids have effects on carbohydrate, fat, and protein metabolism, cardiovascular system, fluid and electrolyte balance, inflammation and immune system, reproduction and embryonic development, mood and cognitive [12].
- Suggestion to rephase: “…mood, embryonic and cognitive development?
Response: This change has been made.
Line 113
Blood samples were collected (with heparin) at 18 different times (0, 0.08, 0.17, 0.25, 0.5, 0.75, 1, 2, 4, 6, 8, 10, 12, 18, 24, 36, and 48 h) and plasma and milk samples were adequately stored at -80 °C until analysis. Validation data is in accordance with criteria.
Response: Thank you for your valuable opinions.
- Nevertheless, neither the storage period nor stability of drug was assayed. Please comment.
- Was photodegradation considered? (protected from light?)
- Response: Blood samples were centrifuged 1 hour after collection and then the plasma samples obtained were stored at -20°C. Milk samples were also stored at -20 oC immediately after collection. Both plasma and milk samples were taken on the same day at -80 oC. Danofloxacin analysis in plasma and milk samples was performed within 3 months.
- Amber microcentrifuge tubes were used to prevent photodegradation during analysis.
- Since the aim of this study was to determine the effect of dexamethasone on plasma and milk disposition kinetics of danofloxacin in lactating sheep, the validation of the method was not presented in detail.
- However, in line with your recommendations, method validation is presented in detail below.
In this study, the stability of danofloxacin was determined. Three freeze–thaws, long-term and post-preparative stabilities of danofloxacin in sheep plasma were investigated using low and high quality control samples and evaluated by the calculated bias between observed and theoretical concentration. The freeze–thaw stability of danofloxacin was determined over three freeze–thaw cycles within 3 days. In each freeze–thaw cycle, the spiked plasma samples were frozen at -80 oC for 24 h and thawed at room temperature. The short-term stability was examined by analyzing triplicates of the frozen low and high QC samples kept at room temperature for 24 h before sample preparation. The long-term stability was evaluated by keeping in triplicate the two concentration levels of routine QC samples with storage at -80 oC for 90 days. The post-preparative stability was evaluated by keeping in triplicate the two extracted QC samples in the autosampler under normal analytical conditions for 24 h. The samples were analyzed and the results were compared with those obtained for the freshly prepared samples. For all stability studies, acceptance criterion was as follows: ±15% bias from theoretical value.
The results of the stability test that includes freeze-thaw, short-term, long-term and post-preparative stabilities were within acceptable limits with ±15% bias and <15% CV.
Line 151
“The definitions and abbreviations of each pharmacokinetic parameter are provided in the footnote of Table 1.”
- This is not necessary.
Response: Thank you for your valuable opinions.
Line 198
In table legend: “a,b,c: Various letters in the same row are statistically significant (p<0.05)”.
- Regarding the statistical significance, clarification of the use “a, b, c” would be helpful.
- Response: The study consisted of 3 groups. Differences between groups were evaluated using ANOVA and Tukey tests. The letters a, b, c show the statistical difference between the groups. We think that it would not be appropriate to state the meaning of these letters separately in this statistical evaluation made between the 3 groups.
Line 280
“However, in sheep, the BCRP inhibitor triclobendazole metabolites did not change the milk pharmacokinetics of danofloxacin;”
- Please consider “triclabendazole” (two items found as triclobendazole)
- Response: This change has been made.
Line 305
Considering resistance global problems and knowing that fluoroquinolones are categorized by WHO as “highest priority critically important antimicrobials” (HPCIA) and by EMA (AMEG category – cat B “Restrit”), further discussion/reference to this topic would be advisable/welcome.
Response: Based on your suggestions, the following sentence was added to the discussion section.
“Fluoroquinolones are crucial antibiotics in terms of antimicrobial resistance risk and are classified as “highest priority critically important antimicrobials” by the World Health Organization and “category B (restriction)” by the European Medicines Agency. Therefore, it is essential to pay attention to the withdrawal time to prevent the development of resistance to these antibiotics [51,52].”

Reviewer 2 Report
Comments and Suggestions for Authors
This manuscript nicely characterizes the effect of dexamethasone on plasma and milk pharmacokinetics of danofloxacin. This study is really very interesting and useful for the field with a high-standard scientific performance. Drug-drug interactions are extremely important regarding the problem of drug residues. In this regards, authors should consider to justify the study of the effect on milk concentration of danofloxacin in the Introduction section and at the beginning of the Discussion, highlighting its relevance for drug residues in milk.
Other minor points that should be addressed are the following:
-2.1. Animals. Please, give more relevant information about the animals used, such as lactation period and milk production
-2.2. Experimental design: Please, give more relevant information about the milk sampling method. It is important to indicate whether milking was performed with complete evacuation of the udder with a milking machine and an aliquot sampling, or just with a manual sampling.
3.2. and 3.3. Plasma and milk pharmacokinetic parameters: As indicated in line 113, sampling points include 36 and 48 hours. However, in Figure 1 only data until 24 hours is reported and in Figure 2, only data until 36 hours. Please, justify.
-Lines 280 and 282: “Triclobendazole”
-Lines 283-284: Please, include reference for dexamethasone as a BCRP substrate
-Lines 293-295: Reason for the lack of effect of dexamethasone in plasma PK of danofloxacin is already adequately hypothesized before (lines 267-269). Therefore, it is not necessary to point to different BCRP sensitivity depending on the tissue because this is a really hard issue.
Author Response
Reviewer 2
This manuscript nicely characterizes the effect of dexamethasone on plasma and milk pharmacokinetics of danofloxacin. This study is really very interesting and useful for the field with a high-standard scientific performance. Drug-drug interactions are extremely important regarding the problem of drug residues. In this regards, authors should consider to justify the study of the effect on milk concentration of danofloxacin in the Introduction section and at the beginning of the Discussion, highlighting its relevance for drug residues in milk.
Response: Thank you for your valuable opinions and suggestions. Information about the importance of drug residues in milk was presented in the discussion section.
Other minor points that should be addressed are the following:
-2.1. Animals. Please, give more relevant information about the animals used, such as lactation period and milk production
Response: This information has been added to the relevant section.
“The experiment included six healthy lactating Akkaraman sheep (milk production 335 ± 25 g/day), averaging 42.57 kg in weight and aged between 2.4 and 2.8 years. The study was conducted on the 30-70 days of lactation.”
The study was designed as a three-period crossover trial (2 × 2 × 2) with at least 20 days of washout between each period. Therefore, the study was conducted on days 30–70 of lactation.
-2.2. Experimental design: Please, give more relevant information about the milk sampling method. It is important to indicate whether milking was performed with complete evacuation of the udder with a milking machine and an aliquot sampling, or just with a manual sampling.
Response: Milk samples were taken by manuel milking at the specified times. Approximately 2 ml of milk sample was taken for each sampling and then the udder was completely emptied.
3.2. and 3.3. Plasma and milk pharmacokinetic parameters: As indicated in line 113, sampling points include 36 and 48 hours. However, in Figure 1 only data until 24 hours is reported and in Figure 2, only data until 36 hours. Please, justify.
Response: Plasma and milk samples were collected for 48 hours. However, danofloxacin was detected in plasma for up to 24 hours and in milk for up to 36 hours in both the single and combined treatment groups. This information was added to sections 3.2 and 3.3.
-Lines 280 and 282: “Triclobendazole”
Response: These changes have been made.
-Lines 283-284: Please, include reference for dexamethasone as a BCRP substrate
Response: This statement that dexamethasone is a bcrp substrate was incorrect. I'm sorry about that. I've deleted this statement.
-Lines 293-295: Reason for the lack of effect of dexamethasone in plasma PK of danofloxacin is already adequately hypothesized before (lines 267-269). Therefore, it is not necessary to point to different BCRP sensitivity depending on the tissue because this is a really hard issue.
Response: This sentence was deleted.
